# YES1 as a Therapeutic Target for HER2-Positive Breast Cancer after Trastuzumab and Trastuzumab-Emtansine (T-DM1) Resistance Development

**DOI:** 10.3390/ijms222312809

**Published:** 2021-11-26

**Authors:** Miwa Fujihara, Tadahiko Shien, Kazuhiko Shien, Ken Suzawa, Tatsuaki Takeda, Yidan Zhu, Tomoka Mamori, Yusuke Otani, Ryo Yoshioka, Maya Uno, Yoko Suzuki, Yuko Abe, Minami Hatono, Takahiro Tsukioki, Yuko Takahashi, Mariko Kochi, Takayuki Iwamoto, Naruto Taira, Hiroyoshi Doihara, Shinichi Toyooka

**Affiliations:** 1Department of General Thoracic Surgery and Breast and Endocrinological Surgery, Graduate School of Medicine Dentistry and Pharmaceutical Sciences, Okayama University, Okayama 700-8558, Japan; p8fk8zyg@s.okayama-u.ac.jp (M.F.); k.shien@okayama-u.ac.jp (K.S.); ksuzawa@okayama-u.ac.jp (K.S.); povc49wk@s.okayama-u.ac.jp (Y.Z.); p2x1815i@s.okayama-u.ac.jp (T.M.); pfgl68ud@s.okayama-u.ac.jp (Y.O.); me19093@s.okayama-u.ac.jp (R.Y.); me19010@s.okayama-u.ac.jp (M.U.); pyv19e5d@s.okayama-u.ac.jp (Y.S.); pct67p3v@okayama-u.ac.jp (Y.A.); p1oz0wy1@s.okayama-u.ac.jp (M.H.); takahirot5974@yahoo.co.jp (T.T.); yukotaka@okayama-u.ac.jp (Y.T.); mariko-leigh-k@md.okayama-u.ac.jp (M.K.); tiwamoto@md.okayama-u.ac.jp (T.I.); ntaira@md.okayama-u.ac.jp (N.T.); hdoihara@md.okayama-u.ac.jp (H.D.); toyooka@md.okayama-u.ac.jp (S.T.); 2Departments of Pharmacy, Okayama University Hospital, Okayama 700-8558, Japan; ph422025@s.okayama-u.ac.jp

**Keywords:** breast cancer, YES1, T-DM1, dasatinib, drug resistance

## Abstract

Trastuzumab-emtansine (T-DM1) is a therapeutic agent molecularly targeting human epidermal growth factor receptor 2 (HER2)-positive metastatic breast cancer (MBC), and it is especially effective for MBC with resistance to trastuzumab. Although several reports have described T-DM1 resistance, few have examined the mechanism underlying T-DM1 resistance after the development of acquired resistance to trastuzumab. We previously reported that YES1, a member of the Src family, plays an important role in acquired resistance to trastuzumab in *HER2*-amplified breast cancer cells. We newly established a trastuzumab/T-DM1-dual-resistant cell line and analyzed the resistance mechanisms in this cell line. At first, the T-DM1 effectively inhibited the *YES1*-amplified trastuzumab-resistant cell line, but resistance to T-DM1 gradually developed. *YES1* amplification was further enhanced after acquired resistance to T-DM1 became apparent, and the knockdown of the *YES1* or the administration of the Src inhibitor dasatinib restored sensitivity to T-DM1. Our results indicate that YES1 is also strongly associated with T-DM1 resistance after the development of acquired resistance to trastuzumab, and the continuous inhibition of YES1 is important for overcoming resistance to T-DM1.

## 1. Introduction

Human epidermal growth factor receptor 2 (HER2)-positive breast cancer accounts for approximately 20% of malignant breast tumors [1]. HER2-positive breast cancer patients used to have poor outcomes, but the recent development of novel drugs based on molecular-targeting mechanisms have improved outcomes for patients with this type of breast cancer. Trastuzumab-emtansine (T-DM1), an antibody-drug conjugate (ADC), is comprised of the humanized, monoclonal, anti-HER2 antibody trastuzumab conjugated via a non-cleavable linker to the anti-microtubule maytansinoid drug emtansine (DM1) [2]. T-DM1 was approved for HER2-positive metastatic breast cancer (MBC) based on the phase Ⅲ EMILIA trial [3] and is usually administered when disease progression is detected after treatment with trastuzumab-containing regimens. In HER2-positive breast cancer patients, HER2-targeted therapies are given sequentially even after acquired resistance to previous regimens has been documented [4]. Therefore, it is advisable to continue treatments considering HER2 expression status, HER2-related signaling, and mechanisms of resistance to previous regimens. There are several reports on the mechanisms responsible for T-DM1 resistance [5,6], but few studies have examined the mechanisms underlying T-DM1 resistance after acquired resistance to trastuzumab develops. We previously established a trastuzumab-resistant breast cancer cell line (named BT-474-R) from the *HER2*-amplified cell line BT-474 and reported that *YES1*, a member of the Src family, plays an important role in acquired resistance to trastuzumab [7]. In the present study, we established a trastuzumab/T-DM1-dual-resistant cell line (named BT-474-R/TDR) and investigated the mechanism accounting for resistance to T-DM1, including whether there was any change in the amplification of *YES1* and the status of HER2-related signaling.

## 2. Results

### 2.1. Establishment of Trastuzumab/T-DM1-Dual-Resistant Cell Line

We previously established a trastuzumab-resistant breast cancer cell line (named BT-474-R) from the *HER2*-amplified cell line BT-474 [8]. In this study, we newly established a trastuzumab/T-DM1-dual-resistant cell line (named BT-474-R/TDR). BT-474-R was additionally treated with increasing doses of T-DM1 (from 0.1 μg/mL to 40 μg/mL). To confirm the resistance to T-DM1, we performed a cell viability assay. The respective IC_50_ values of BT-474, BT-474-R and BT-474-R/TDR for trastuzumab were 0.37 μg/mL, >1000 μg/mL and >1000 μg/mL (Figure 1A), and for T-DM1, they were 0.38 μg/mL, 13.03 μg/mL and >1000 μg/mL (Figure 1B). The trastuzumab-resistant cell line, BT-474-R, was initially sensitive to T-DM1 but gradually acquired resistance to it. As for the sensitivity to DM1-CH_3_, the IC_50_ values were 126.02 nM, 111.26 nM and >1000 nM, respectively (Figure 1C). BT-474 and BT-474-R showed the same level of sensitivity to DM1-CH_3_, suggesting that *YES1* amplification had no effect on sensitivity to DM1-CH_3_. In contrast, BT-474-R/TDR had also acquired resistance to DM1-CH_3_.

### 2.2. HER2-Related Signaling Status

We evaluated the expressions and the phosphorylation levels of the HER2-related signaling molecules in the BT-474, BT-474-R and BT-474-R/TDR cell lines (Figure 2). The HER2 protein expression levels and *HER2* amplification did not differ among these cell lines (Appendix A). Both BT-474-R and BT-474-R/TDR showed elevated HER2 and Akt phosphorylations. The phosphorylation of MAPK was downregulated in BT-474-R/TDR.

### 2.3. YES1 Amplification and the Effects of the YES1 Knockdown in BT-474-R and BT-474-R/TDR

We focused on YES1 based on our previous study [7]. In Western blotting experiments, the phosphorylation levels of Src and YES1 expression were upregulated in BT-474-R/TDR as well as BT-474-R (Figure 2). We also determined the copy number of *YES1*. *YES1* was amplified in BT-474-R and further amplified in BT-474-R/TDR (Figure 3A). To assess the impact of *YES1* amplification on T-DM1 resistance, the siRNA-mediated suppression of YES1 expression was examined in BT-474, BT-474-R and BT-474-R/TDR. The efficacy of the *YES1* knockdown was confirmed by Western blotting (Figure 3B). The expression of YES1 and the phosphorylation of Src were inhibited in BT-474-R and BT-474-R/TDR. Next, we confirmed significant recovery of sensitivities to T-DM1 after the knockdown of *YES1* in BT-474-R and BT-474-R/TDR (Figure 3C).

### 2.4. Effects of the Src Inhibitor Dasatinib in BT-474-R and BT-474-R/TDR

We tested whether the Src inhibitor dasatinib restores sensitivity to T-DM1 employing a cell viability assay (Figure 4A) and a colony formation assay (Figure 4B). In both assays, the combination of T-DM1 with dasatinib was found to be effective in BT-474-R and BT-474-R/TDR. The effects of dasatinib alone and the combination therapy were more marked in the colony formation assay with a longer drug exposure time than in the cell viability assay. We also performed Western blotting to assess the impact of combination therapy on the HER2-related signaling pathway (Figure 4C). In BT-474-R and BT-474-R/TDR, the phosphorylations of HER2 and Src were downregulated by exposure to dasatinib. The phosphorylation of Akt was inhibited by the combination therapy. As shown in Figure 4D, we evaluated the amount of cleaved PARP (poly (ADP-ribose) polymerase), an apoptosis marker. In BT-474 and BT-474-R, which have sensitivity to T-DM1, apoptosis was induced by exposure to T-DM1 alone and by the combination therapy. In BT-474-R/TDR, there was no apoptosis with dasatinib alone, but apoptosis was induced by the combination therapy.

## 3. Discussion

The development of anti-HER2 agents for breast cancer has been remarkable. T-DM1 is a novel ADC targeting HER2-positive MBC that can overcome resistance to trastuzumab [2]. The phase Ⅲ EMILIA study [3] evaluated the safety and efficacy of T-DM1 compared with lapatinib plus capecitabine for HER2-positive patients with locally advanced breast cancer and MBC previously treated with trastuzumab and a taxane agent. T-DM1 produced a significant improvement in both progression-free survival and overall survival, and rates of ≥grade 3 adverse events were lower with T-DM1 than with lapatinib plus capecitabine. In addition, trastuzumab deruxtecan (T-Dxd), a new ADC, was approved for the treatment of breast cancer patients who had developed resistance to trastuzumab and T-DM1 [9,10]. In the current clinical setting, the standard strategies for HER2-positive MBC are trastuzumab, pertuzumab plus a taxane as first-line therapy, followed by T-DM1 and other anti-HER2 agents, such as T-Dxd or lapatinib plus capecitabine [4]. T-Dxd is highly effective for HER2-positive MBC that has become resistant to trastuzumab or T-DM1, but this drug is associated with severe adverse events, such as interstitial pneumonia. Since there are few severe adverse events with T-DM1, and this agent can be used for any HER2-positive MBC, including those in elderly patients, and there is no solid evidence supporting the efficacy of agents other than T-Dxd, it is clinically advisable to overcome T-DM1 resistance and endeavor to sustain its efficacy.

T-DM1 is comprised of the anti-HER2 antibody trastuzumab and the cytotoxic agent DM1. The mechanisms by which resistance to T-DM1 develops are complex and have been widely investigated. Possible mechanisms include the suppression of trastuzumab-mediated effects, impairment of DM1-mediated cytotoxicity and the intracellular trafficking and metabolism of T-DM1 due to the unique ADC form of T-DM1 [5,6]. Although the mechanisms of T-DM1 resistance development have been well studied, there are few reports on the mechanisms underlying resistance to T-DM1 following the acquisition of trastuzumab resistance along the clinical course. In the sequential treatment of HER2-positive MBC, it is useful to consider the mechanisms by which resistance to previous regimens developed.

In this study, we established a trastuzumab/T-DM1-dual-resistant cell line and showed that one of the mechanisms underlying resistance to T-DM1, acquired after trastuzumab resistance developed, was further amplification of *YES1* (Figure 3). The Src family includes c-Src, YES1, Fyn, Lyn, Fgr, Blk, Hck, Lck and Frk [11]. The elevated Src protein levels have been found in many cancers, including those of the colon, breast, pancreas and brain [12]. Src is activated through a variety of mechanisms involving interactions with receptor tyrosine kinases (RTK), such as EGFR and HER2, other signaling pathways such as PIK3K/Akt, Ras/MAPK and Stat3 and integrin receptors [13,14,15,16,17,18]. Specifically, activated Src can bind to the cytoplasmic tail of HER2 and facilitate the formation of heterocomplexes, and it can then phosphorylate tyrosine residues that may lead to the activation of HER2 receptors independently of the ligand [12]. Our previous study revealed that among Src family members, YES1 also binds to HER2 [19]. These interactions enable metastatic processes, including cell proliferation, decreased apoptosis, invasion, migration, adhesion and angiogenesis, to occur.

We found that the phosphorylations of HER2 and Akt were upregulated in resistant cell lines, as was the phosphorylation of Src (Figure 2), and that these upregulations were suppressed by dasatinib due to the interaction described above (Figure 4C). These results were consistent with those reported previously [7,13]. In contrast, the phosphorylation of MAPK was downregulated in BT-474-R/TDR, suggesting that the MAPK signaling pathway might be inhibited by the negative feedback due to the activation of the PI3K/Akt pathway. In our previous study [7], the phosphorylation of MAPK was also shown to be downregulated in BT-474-R. The difference between the results of our present and prior studies indicates that MAPK signaling might be highly susceptible to changes in other signaling pathways.

As for *YES1* amplification, the enhancement was greater in BT-474-R/TDR than in BT-474-R (Figure 3A), and the combined treatment with dasatinib was effective even after resistance was acquired to both trastuzumab and T-DM1 (Figure 4A,B). In addition, prolonged exposure was associated with high efficacies of dasatinib monotherapy and combination therapy (Figure 4A,B). These results suggest that the mechanism by which resistance develops to trastuzumab is also very likely to be involved in T-DM1 resistance and that the continuous inhibition of YES1 may be important for overcoming resistance to T-DM1. We also performed a siRNA-mediated knockdown of c-Src, which is the predominant member of the Src family, but the effects of T-DM1 in BT-474-R and BT-474-R/TDR were not recovered after the knockdown of c-Src (Appendix A). HER2-positive MBC is often treated with trastuzumab-containing regimens, such that continued inhibition of trastuzumab resistance mechanisms might be beneficial in clinical practice. We have found dasatinib to play an important role in overcoming resistance to trastuzumab and T-DM1 in breast cancer cell lines. Dasatinib is known to be an Src inhibitor, and it also targets YES1 [20]. Dasatinib was approved for the treatment of chronic myelocytic leukemia as well as for treating Philadelphia chromosome-positive acute lymphatic leukemia [21]. The possibility that YES1 might serve as a therapeutic target for breast cancer therapy as a means of overcoming drug resistance has already been reported. Previous studies showed the role of *YES1* amplification in T-DM1 [22], trastuzumab and lapatinib [7], and neratinib [19] regimens. Among members of the Src family, *YES1* expression showed the highest association with poor outcomes in patients with non-small cell lung cancer [23]. We also previously reported [7] that higher YES1 expression correlated with poor outcomes for HER2-positive breast cancer patients, and the inhibition of YES1 appears to be an urgent clinical issue in breast cancer and other malignancies. There have been clinical trials that focused on dasatinib targeting breast cancers. The efficacy and safety of dasatinib with trastuzumab and paclitaxel as first-line therapy for HER2-positive MBC have been reported [24]. It was concluded that dasatinib could be safely combined with trastuzumab and paclitaxel and that the combination therapy is effective with an objective response rate of almost 80%. Another report described the effect of dasatinib for hormone receptor-positive breast cancer patients with resistance to hormonal therapy [25]. Thus, dasatinib might be useful for HER2-positive and other types of breast cancer. Additional clinical trials are needed, and the clinical use of dasatinib for breast cancer is anticipated.

In addition to the amplification of *YES1*, the reduced sensitivity to DM1 is apparently associated with T-DM1 resistance (Figure 1C), but the mechanism of acquired resistance to DM1 was not determined in our present study. The protein expression levels of epithelial-mesenchymal transition markers such as E-cadherin and N-cadherin, cancer stem cell markers such as vimentin and ALDH1 and ATP-binding cassette (ABC) transporters such as ABCB1 and ABCG2, which are known to be involved in mechanisms of resistance development to cytotoxic agents, were not changed (data not shown). Previous reports [5,6] noted that the mechanisms of impairment of DM-1-mediated cytotoxicity include increased expressions of drug efflux transporters [26,27,28,29], mutations in tubulin [30], escape from mitotic catastrophe through reduced induction of cyclin B1 and increased expression of polo-like kinase 1 [31,32,33]. Further studies are needed to elucidate the mechanisms underlying the reduced sensitivity to DM1 in our resistant cell line.

In conclusion, T-DM1 was effective against the *YES1*-amplified trastuzumab-resistant cell line, and further amplification of *YES1* was shown to be associated with the development of T-DM1 resistance. Furthermore, the knockdown of *YES1* or the administration of dasatinib is effective even after the development of acquired resistance to both trastuzumab and T-DM1. The mechanisms of resistance to trastuzumab are highly involved in T-DM1 resistance, and the continuous inhibition of YES1 thereby plays an important role in overcoming resistance to T-DM1.

## 4. Materials and Methods

### 4.1. Cell Culture and Drugs

The HER2-positive breast cancer cell line BT-474 (catalog number: HTB-20) was used in this study. This cell line was purchased from the American Type Culture Collection (Manassas, VA, USA). We previously established a trastuzumab-resistant cell line, BT-474-R, by treating BT-474 with increasing doses of trastuzumab (from 0.1 μg/mL to 40 μg/mL) for 10 months [8]. BT-474-R was additionally treated with a starting dose of 0.1 μg/mL, which was determined by the IC_10_ values and generated by continuous exposure to increasing doses of T-DM1 up to 40 μg/mL for 12 months. The cells were exposed to T-DM1 until they were damaged at the 30% confluence of the dish, and they were then passaged when they reached 80% confluence in a drug-free state. When the cells reached 80% confluence, the drug was exposed again. The cells were repeatedly treated with the same concentration of T-DM1 until almost all of the cells survived the treatment. The cells were cultured in Dulbecco’s Modified Eagle Medium supplemented with 10% fetal bovine serum and maintained under 5% CO_2_ at 37 °C. Trastuzumab and T-DM1 were purchased from Chugai Pharmaceutical Co., Ltd. (Tokyo, Japan), dasatinib was purchased from the Bristol-Myers Squibb Company (New York, NY, USA) and DM1-CH_3,_ which is the methyl form of DM1, from Toronto Research Chemicals (Toronto, ON, Canada).

### 4.2. Cell Viability Assay

The MTS assay was performed to measure antitumor effects. The cells were equally seeded on a 96-well plate (3000 cells/well), and drug dilutions were added 24 h later. After 72 h, the cell proliferation was determined using a CellTiter 96 AQueous bromide One Solution Reagent (Promega, Fitchburg, WI, USA). The optical densities of the samples were measured at 492 nm using a Multiskan^TM^ FC Microplate Photometer (Thermo Fisher Scientific, Waltham, MA, USA). The IC_50_ value, which refers to the concentration of the drug required to inhibit cell proliferation by 50%, was calculated based on the results of the MTS assay. All experiments were repeated three times. The data were expressed as the mean ± standard error (SE).

### 4.3. Western Blotting

The total cell lysate was extracted with lysis buffer, a mixture of RIPA buffer, phosphatase inhibitor cocktails 2 and 3 (Sigma-Aldrich, St. Louis, MO, USA) and Complete Mini (Roche, Basel, Switzerland). The extracted protein was quantitated using a protein assay (Bio-Rad Laboratories) and then transferred to a membrane using a transfer system (Bio-Rad Laboratories, Hercules, CA, USA). The membrane was then blocked with 5% skim milk in TBC-containing 0.05% Tween 20 (T-TBS) for 1 h. The membranes were probed with primary antibodies diluted with 5% bovine serum albumin overnight at 4 °C, followed by incubation with the secondary antibody for 1 h at 25 °C. To detect specific signals, we examined the membrane using the ECL Prime Western Blotting Detection System (GE Healthcare, Amersham, UK) and the LAS-3000 (Fuji Film, Tokyo, Japan). The primary antibodies were as follows: phospho-HER2 (Tyr877), HER2, phospho-Src family (Tyr416), YES1, phospho-Akt (Ser473), Akt, phospho-MAPK (Tyr202/204), MAPK, cleaved PARP (Cell Signaling Technology, Beverly, MA, USA ) and Actin (used as the loading control) (Merck Millipore, Darmstadt, Germany). The secondary antibodies were HRP-conjugated anti-mouse or anti-rabbit IgG (Santa Cruz Biotechnology, Santa Cruz, CA, USA).

### 4.4. Copy Number Assay

The DNA was isolated using a DNeasy Blood and Tissue Kit (Qiagen, Venlo, Netherlands). The copy number was determined with the StepOnePlus Real-Time PCR System (Thermo Fisher Scientific) using TaqMan copy number assays (Thermo Fisher Scientific). A TaqMan RNase P Control (Thermo Fisher Scientific) was used as the reference gene. The relative copy number in each sample was determined by comparing the ratio of the Ct value of the target gene to that of the reference gene in each sample with the ratio in standard genomic DNA (Merck). The assays were repeated three times. The data were expressed as the mean and SE.

### 4.5. siRNA Transfection

The small interfering RNAs (siRNAs) specific for YES1 (Silencer^®^ Select Validated siRNA #4390824) and the non-targeting control (Silencer^®^ Select Negative Control No.2 siRNA #4390846) were purchased from Thermo Fisher Scientific. The cells were reverse-transfected with 10 nM siRNAs mixed with Lipofectamine^®^ RNAiMAX Transfection (Thermo Fisher Scientific). After the siRNA transfection, the cells were incubated for 72 h under 5% CO_2_ at 37 °C.

### 4.6. Colony Formation Assay

The cells were seeded in 6-well plates (1000 cells/well) in triplicate and incubated under 5% CO_2_ at 37 °C. After their attachment to the plates, the cells were treated with T-DM1 (1 μg/mL) alone, dasatinib (100 nM) alone or the combination of these two agents for 14 days. The culture solution was removed, and the plates were washed twice with phosphate-buffered saline. The colony was fixed employing 4% formaldehyde for 20 min and stained using 0.2% crystal violet for 5 min. The colony counts were measured using ImageJ version 1.53 (National Institutes of Health, Bethesda, MD, USA). The assays were repeated three times. The data were expressed as means and SE.

### 4.7. Statistical Analysis

All statistical analyses were carried out with JMP^®^ 14 (SAS Institute Inc., Cary, NC, USA). The results of the comparisons between groups were statistically assessed using the *t*-test (two groups) or the one-way analysis of variance (multiple groups). Values of *p* less than 0.05 were considered to indicate a statistically significant difference.

## Figures and Tables

**Figure 1 ijms-22-12809-f001:**
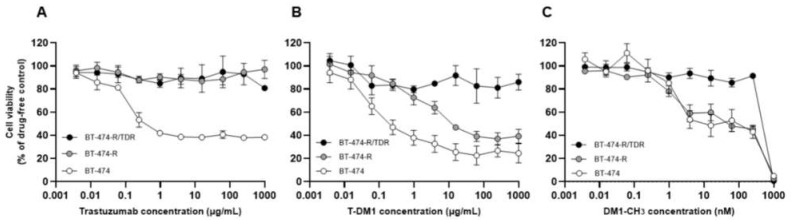
The establishment of trastuzumab-resistant cell line BT-474-R and trastuzumab/T-DM1-dual-resistant cell line BT-474-R/TDR. MTS assay to assess sensitivity to (**A**) Trastuzumab, (**B**) T-DM1 and (**C**) DM1-CH_3_ in BT-474, BT-474-R and BT-474-R/TDR. These assays were repeated three times. The data are shown as means ± SE.

**Figure 2 ijms-22-12809-f002:**
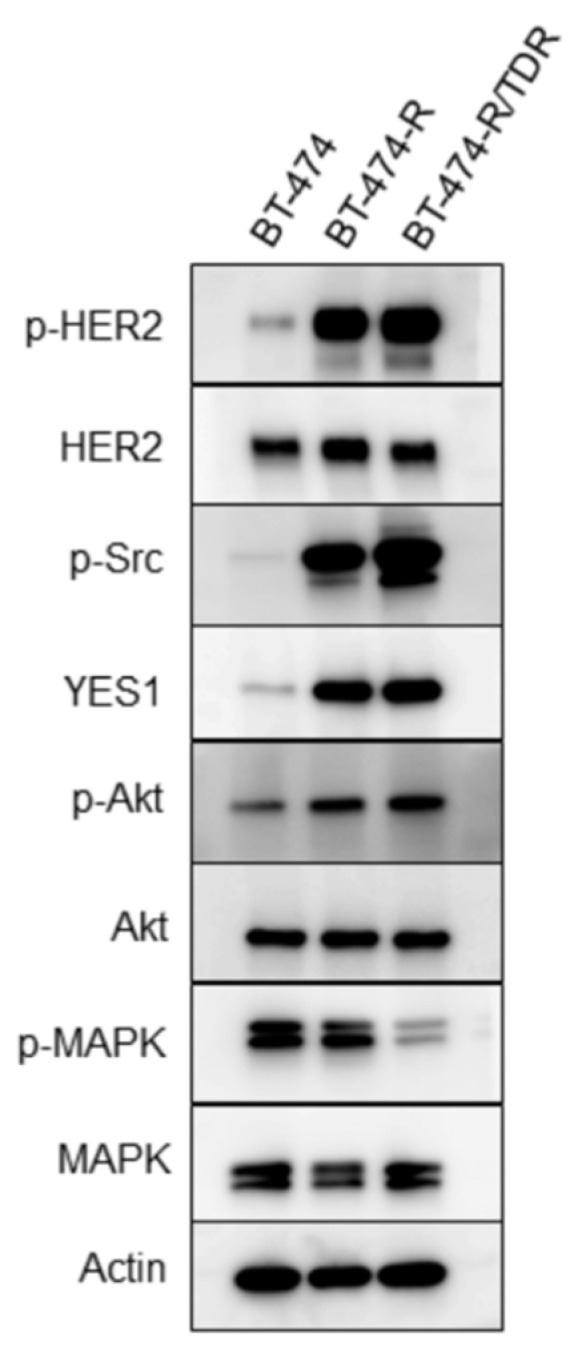
The protein expression and phosphorylation levels of the HER2-related signaling molecules in BT-474, BT-474-R and BT-474-R/TDR.

**Figure 3 ijms-22-12809-f003:**
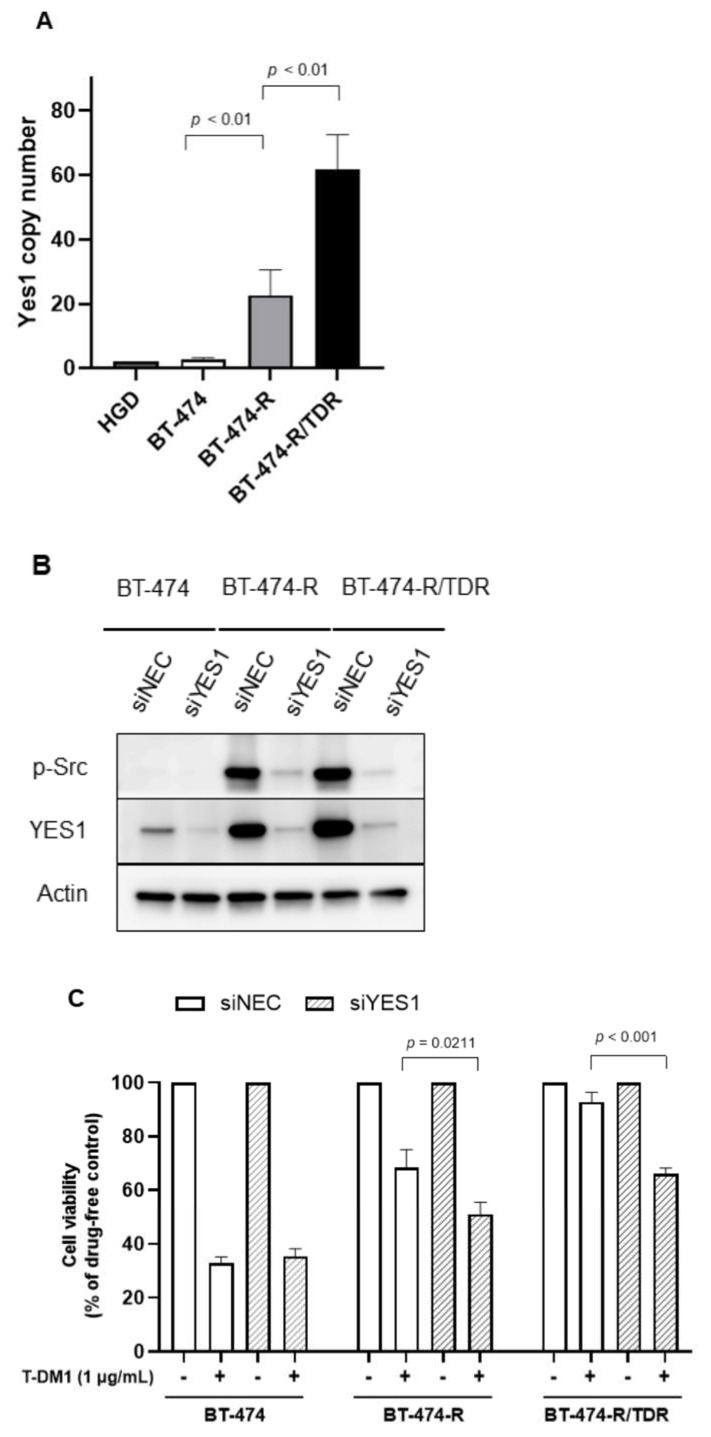
*YES1* amplification in BT-474-R and BT-474-R/TDR. (**A**) The copy number assay of *YES1*. *YES1* was amplified in BT-474-R and further amplified in BT-474-R/TDR. Human Genomic DNA (HGD) was used as a control (2 copies). The assay was repeated three times. The data are shown as means + SE. (**B**) The phosphorylations of Src and YES1 after the *YES1* knockdown by *YES1* siRNA in BT-474, BT-474-R and BT-474-R/TDR. (**C**) Drug sensitivities to T-DM1 after the *YES1* knockdown are calculated employing the MTS assay in BT-474, BT-474-R and BT-474-R/TDR. The assay was repeated three times. The data are shown as means + SE.

**Figure 4 ijms-22-12809-f004:**
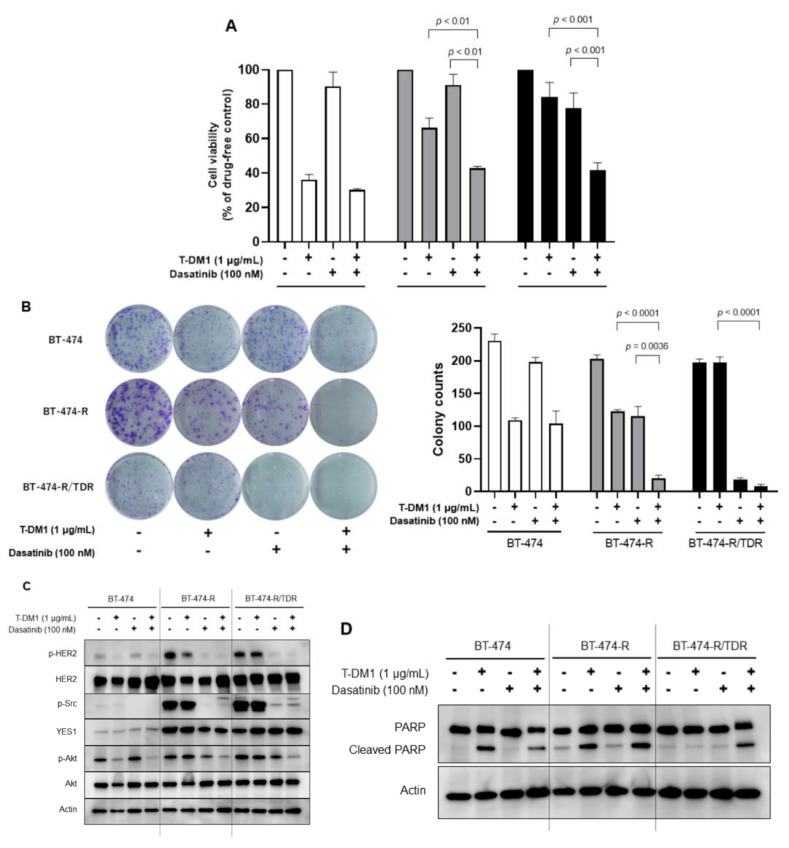
The combination of T-DM1 plus the Src family inhibitor dasatinib was effective against BT-474-R and BT-474-R/TDR. (**A**) Drug sensitivities to T-DM1 and/or dasatinib are calculated employing the MTS assay in BT-474, BT-474-R and BT-474-R/TDR. (**B**) Colony formation assay to assess the effect of prolonged exposure to T-DM1 and/or dasatinib. The assay was repeated three times. The data are shown as means + SE. (**C**) Effects of T-DM1 and/or dasatinib on protein phosphorylations in BT-474, BT-474-R and BT-474-R/TDR. The cells were treated with the drugs for 2 h. (**D**) Apoptosis assay for the expressions of PARP and cleaved PARP in BT-474, BT-474-R, and BT-474-R/TDR. The cells were treated with T-DM1 alone, dasatinib alone or the T-DM1 plus dasatinib combination for 48 h.

## Data Availability

Data will be available if requested.

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
