# Peer review of "YES1 as a Therapeutic Target for HER2-Positive Breast Cancer after Trastuzumab and Trastuzumab-Emtansine (T-DM1) Resistance Development"

_ijms, 2021, doi:10.3390/ijms222312809_

Round 1

Reviewer 1 Report

The authors established and analysed a Trastuzumab/T-DM1-dual resistant cell line. Data shown here indicate a potential role of Yes-1 in the underlying resistance. Additional experiments should be performed to gain more insights into the resistance mechanism.

- Yes1 amplification is observed at DNA level (Figure 3A), however, no increase in YES1 protein levels is found (Figure 3B). How can these results be reconciled? Is YES1 DNA amplification required for resistance to TDR?

- Is HER2 also further amplified in TDR-resistant cells compared to the parental cells?

- Figure 3C: Similar experiments should be conducted c-SRC knocked-down cells

- Why did the authors choose a T-DM1 concentration of 1 microg/mL in all the combination experiments?

- Figure 4: Are these data recapitulated by combining T-DM1 with an Akt inhibitor?

- Figure 4B: BT-474-R/TDR cells look less proliferating than BT-474-R (-/- condition). Might this explain in part their resistance to T-DM1, which might preferentially target highly cycling cells?

- Can the authors please add more details regarding the generation of the TDR-resistant line in the Methods section?

- Have the authors tried to induce resistance in BT-474-R cells with concomitant Dasatinib treatment? If so, did they observe a delay in resistance onset or no onset of resistance at all? Including these data would be beneficial.

Author Response

We appreciate the reviewers for their helpful comments, and have provided a point-by-point response below. Sections of the manuscript that have been significantly revised are highlighted in yellow, including additional data.

Reviewer #1

The authors established and analyzed a Trastuzumab/T-DM1-dual resistant cell line. Data shown here indicate a potential role of Yes-1 in the underlying resistance. Additional experiments should be performed to gain more insights into the resistance mechanism.

  1. Yes1 amplification is observed at DNA level (Figure 3A), however, no increase in YES1 protein levels is found (Figure 3B). How can these results be reconciled? Is YES1 DNA amplification required for resistance to TDR? 

As the reviewer 1 mentioned, contrary to the results of copy number assay, the protein expression of YES1 was not higher in BT-474-R/TDR than in BT-474-R in Fig. 3B and also in Fig.2. There are two possibilities explaining this discrepancy: the first is that DNA copy number may not necessarily correlate with protein expression or mRNA levels in several types of genes (Myhre S. et al, Mol Oncol 7 (2013) 704-718). The second is that the protein expression levels of YES1 in BT-474-R and BT-474-R-TDMR may have reached saturation limit detected in western blotting. These findings may not affect the result that YES1 is associated with T-DM1 resistance because knockdown of YES1 restored the sensitivity to T-DM1 (Fig. 3C).

  1. Is HER2 also further amplified in TDR-resistant cells compared to the parental cells?

The protein expression levels of HER2 were almost same in BT-474, BT-474-R and BT-474-R/TDR (Fig. 2). Similarly, HER2 copy number was almost same in these three cell lines and there was no further amplification in BT-474-R/TDR, and no significant difference was observed among these cell lines. We added this result as Supplemental Figure 1.

  1. Figure 3C: Similar experiments should be conducted c-SRC knocked-down cells

We appreciate the reviewer’s important suggestion. As the reviewer 1 pointed out, the possibility that c-Src may be involved in T-DM1 resistance cannot be denied because Dasatinib inhibit Src family including c-Src, YES1 and other members. On the other hand, in our previous study (Takeda T. et al, PLoS One 12 (2017) e0171356), c-Src was not amplified in BT-474-R, and YES1 amplification was strongly associated with drug resistance in BT-474-R. According to these findings, we focused on YES1 amplification in this study, and showed that YES1 suppression could overcome T-DM1 resistance in our cells.

  1. Why did the authors choose a T-DM1 concentration of 1 microg/mL in all the combination experiments?

In clinical use, the recommended dose of T-DM1 for breast cancer patients is 3.6mg/kg intravenously every 3 weeks. At this concentration, Cmax is 82.0 ± 10.0 μg/mL, and the trough concentration in blood serum after repeated administration was around 1 μg/mL. In addition, the T-DM1 concentration of 1 μg/mL was determined based on other previous papers (Lei Wang. et al, Br J Cancer 123 (2020) 1000-1011).

  1. Figure 4: Are these data recapitulated by combining T-DM1 with an Akt inhibitor?

As the reviewer suggested, the combination with Akt inhibitor may be effective because the protein upregulation of p-Akt was shown. However, the upregulation of p-Akt could be caused by the upregulation of p-Src, which exists upstream of Akt, so inhibiting Src would be more important than inhibiting Akt. Furthermore, the combination with Dasatinib, which is already in clinical use, would be more meaningful than combination with AKT inhibitors, which are still undergoing clinical trials.

  1. Figure 4B: BT-474-R/TDR cells look less proliferating than BT-474-R (-/- condition). Might this explain in part their resistance to T-DM1, which might preferentially target highly cycling cells?

As the reviewer pointed out, the less proliferation in BT-474-R/TDR than BT-474-R could be involved in acquired T-DM1, especially for DM1 resistance. However, we have not performed analysis including cell cycle, and this point is still unknown. This will be the subject of our future research. We appreciate the reviewer’s important suggestion.

  1. Can the authors please add more details regarding the generation of the TDR-resistant line in the Methods section?

More details such as how to determine the starting concentration and the timing of drug administration were added in the method section, as described below.

The HER2-positive breast cancer cell line BT-474 (catalog number: HTB-20) was used in this study. This cell line was purchased from American Type Culture Collection (Manassas, VA, USA). We previously established a trastuzumab resistant cell line, BT-474-R, by treating BT-474 with increasing doses of trastuzumab (from 0.1 μg/mL to 40 μg/mL) for 10 months [8]. BT-474-R was additionally treated with starting dose of 0.1 μg/mL, which was determined by the IC10 values and generated by continuous exposure to increasing doses of T-DM1 up to 40 μg/mL for 12 months. The cells were exposed to T-DM1 until cells were damaged at the 30% confluence of the dish, and then passaged when they reached at 80% confluence in drug free state. When the cells reached 80% confluence, drug was exposed again. The cells were repeatedly treated with the same concentration of T-DM1 until almost all of the cells survived the treatment. The cells were cultured in Dul-becco’s modified Eagle medium supplemented with 10% fetal bovine serum and maintained under 5% CO2 at 37℃. 

  1. Have the authors tried to induce resistance in BT-474-R cells with concomitant Dasatinib treatment? If so, did they observe a delay in resistance onset or no onset of resistance at all? Including these data would be beneficial.

Thank you for the reviewer’s important comments. We have not attempted to induce resistance in BT-474-R with concomitant Dasatinib, and we will definitely try them.

Reviewer 2 Report

The authors seek to explore targeting YES1 in resistant HER2+ breast cancer.  Understanding more about how resistance occurs and may be overcome is an important unmet medical need.  They tested T-DM1 treatment in a trastuzumab resistant cell line and were able to develop dual resistance.  Yes1 was amplified and knockdown of Yes1 or treating with the Src/Yes1 inhibitor dasatinib, restored T-DM1 activity.  This outcome is interesting and contributes to the field.  The authors cite some work on dasatinib in the clinic in combination with trastuzumab, so this particular result may not be fully surprising, but it is nice to have the data to support the theory as to why dasatinib may work in these cases to limit the development/impact of resistance.

The study is well laid out, clearly written, with appropriate replicates, error bars and statistics.  The Western blot results are clear. 

Round 2

Reviewer 1 Report

Regarding point 1, the authors raised the possibility that YES1 protein levels might have reached saturation limit detected in western blotting. Shorter exposure times, or blots with reduced amount of loaded protein extracts or more diluted anti-YES1 antibodies should be included in the manuscript.

  1. Yes1 amplification is observed at DNA level (Figure 3A), however, no increase in YES1 protein levels is found (Figure 3B). How can these results be reconciled? Is YES1 DNA amplification required for resistance to TDR? 

As the reviewer 1 mentioned, contrary to the results of copy number assay, the protein expression of YES1 was not higher in BT-474-R/TDR than in BT-474-R in Fig. 3B and also in Fig.2. There are two possibilities explaining this discrepancy: the first is that DNA copy number may not necessarily correlate with protein expression or mRNA levels in several types of genes (Myhre S. et al, Mol Oncol 7 (2013) 704-718). The second is that the protein expression levels of YES1 in BT-474-R and BT-474-R-TDMR may have reached saturation limit detected in western blotting. These findings may not affect the result that YES1 is associated with T-DM1 resistance because knockdown of YES1 restored the sensitivity to T-DM1 (Fig. 3C).

Regarding point 3, the effects of Dasatinib appear to be stronger than YES1 knock-down. Thus, inclusion of similar c-SRC knock-down experiments would be very important.

  1. Figure 3C: Similar experiments should be conducted c-SRC knocked-down cells

We appreciate the reviewer’s important suggestion. As the reviewer 1 pointed out, the possibility that c-Src may be involved in T-DM1 resistance cannot be denied because Dasatinib inhibit Src family including c-Src, YES1 and other members. On the other hand, in our previous study (Takeda T. et al, PLoS One 12 (2017) e0171356), c-Src was not amplified in BT-474-R, and YES1 amplification was strongly associated with drug resistance in BT-474-R. According to these findings, we focused on YES1 amplification in this study, and showed that YES1 suppression could overcome T-DM1 resistance in our cells.
